# Synthesis of Full-Length cDNA Infectious Clones of *Soybean Mosaic Virus* and Functional Identification of a Key Amino Acid in the Silencing Suppressor Hc-Pro

**DOI:** 10.3390/v12080886

**Published:** 2020-08-13

**Authors:** Wenhua Bao, Ting Yan, Xiaoyi Deng, Hada Wuriyanghan

**Affiliations:** 1Key Laboratory of Herbage and Endemic Crop Biotechnology, School of Life Sciences, Inner Mongolia University, Hohhot 010070, China; nmBWH530236@163.com (W.B.); nmgyanting@163.com (T.Y.); qq1441286922@163.com (X.D.); 2State Key Laboratory of Reproductive Regulation and Breeding of Grassland Livestock School of Life Sciences, Inner Mongolia University, Hohhot 010070, China

**Keywords:** *Potyviridae*, soybean mosaic virus, cDNA infectious clone, SMV-GFP, seed transmission, Hc-Pro, FRNK motif, key amino acid, oligomerization

## Abstract

*Soybean mosaic virus* (SMV), which belongs to the *Potyviridae*, causes significant reductions in soybean yield and seed quality. In this study, both tag-free and reporter gene *green fluorescent protein* (*GFP*)-containing infectious clones for the SMV N1 strain were constructed by Gibson assembly and with the yeast homologous recombination system, respectively. Both infectious clones are suitable for agroinfiltration on the model host *N. benthamiana* and show strong infectivity for the natural host soybean and several other legume species. Both infectious clones were seed transmitted and caused typical virus symptoms on seeds and progeny plants. We used the SMV-GFP infectious clone to further investigate the role of key amino acids in the silencing suppressor helper component-proteinase (Hc-Pro). Among twelve amino acid substitution mutants, the co-expression of mutant 2—with an Asparagine→Leucine substitution at position 182 of the FRNK (Phe-Arg-Asn-Lys) motif—attenuated viral symptoms and alleviated the host growth retardation caused by SMV. Moreover, the Hc-Prom2 mutant showed stronger oligomerization than wild-type Hc-Pro. Taken together, the SMV infectious clones will be useful for studies of host–SMV interactions and functional gene characterization in soybeans and related legume species, especially in terms of seed transmission properties. Furthermore, the SMV-GFP infectious clone will also facilitate functional studies of both virus and host genes in an *N. benthamiana* transient expression system.

## 1. Introduction

The *Potyviridae* represents one of the largest families of plant viruses that cause severe yield losses in many major crops [1,2,3]. *Soybean mosaic virus* (SMV) is a monopartite positive-sense ssRNA virus in the *Potyviridae* with a genome of approximately 10 kb. SMV encodes 11 putative proteins, including potyvirus 1 (P1), helper component-proteinase (Hc-Pro), potyvirus 3 (P3), N-terminal half of P3 fused to PIPO (P3N-PIPO), 6 kinase 1 (6K1), cylindrical inclusion (CI), 6 kinase 2 (6K2), Viral genome-linked Protein (VPg), nuclear inclusion a proteinase (NIa-Pro), nuclear inclusion b (NIb) and coat protein (CP) [4,5]. The production of individual mature proteins is mediated by proteinase activity and is indispensable for its infection. SMV causes obvious phenotypic changes including leaf discoloration, mosaic, curling and dwarfing, and it further leads to severe symptoms such as necrosis and seed mottle [6,7].

As for the SMV isolates, seven strains (G1–G7) were reported in the United States and five strains (A-E) were identified in Japan [8,9,10]. SMV is divided into 21 strains in China, including the northeast N1, N2 and N3 strains, among which N1 is a mild strain, N3 is a virulent strain and N2 is a moderately virulent one [11,12,13]. SMV can be transmitted by aphids and seeds, and the seed transmission rates vary from 0% to 64%, depending on the virus isolates and host genotypes [14,15]. SMV-infected soybean seeds usually show seed mottling characteristics on the epidermis. Seed mottling is controlled by the alleles at the *I* locus, which regulate the accumulation of chalcone synthase (*CHS*) mRNA levels by RNA silencing [16]. In healthy soybeans, the *I* locus allele contains inverted repeats of the *CHS* gene that produce dsRNAs and lead to the tissue-specific silencing of *CHS* expression, resulting in normal yellow seed coats. SMV-infected soybean seeds show brown-mottled symptoms, with elevated *CHS* mRNA levels, which is attributed to the partial suppression of the RNA silencing of the *CHS* mRNAs by SMV Hc-Pro [17]. In addition to Hc-Pro, the P1 and CP cistrons of SMV also participate in seed transmission [14]. The seed infection level may also be one of the indicators of disease susceptibility. Due to its limited infection of alternate host species, seedborne infections are the main sources of primary inoculum for SMV infections [18]. The virus-laden seeds can be stored at low temperature for a long time. For example, *Tobacco Ring Spot Virus* (TRSV) remained on soybean seeds at 2 °C for 5 years [19].

Infectious clones are useful reverse genetic manipulation tools for studies of host–virus interactions and understanding virus infection cycles and pathogenesis [20]. Since the establishment of the first infectious clone for *Brome mosaic virus* (BMV) [21], a variety of plant RNA viruses have been harnessed to generate infectious clones that are transcriptionally controlled by the T7, T3 or 35S promoter [20,22]. Conventional digestion–ligation methods for the construction of infectious clones are usually time consuming and laborious, because they generally require tedious sub-cloning and sequential cloning steps. The development of infectious clones is difficult and inefficient for some viruses because of the large genome size, toxicity to *E.coli* growth and unexpected deletions [23,24,25]. To avoid these problems, novel cloning methods have been used to construct infectious clones. Homologous recombination (HR) in yeast (*Saccharomyces cerevisiae*) is a simple, rapid and efficient way to obtain infectious clones of potyviruses [22]. The method utilizes an efficient homologous recombination system in yeast cells to achieve the assembly of multiple DNA fragments, as 20–30 bp homologous ends can direct yeast endogenous machinery to accurately assemble at least 5–8 recombinant fragments [26,27]. 

Plants evolved sophisticated antiviral mechanisms, among which RNA interference (RNAi) is a common strategy for resisting RNA viruses in a sequence-specific manner [28,29]. To establish successful infections, plant viruses developed RNA silencing suppressors (RSSs) to counteract the host RNAi pathway [30]. Hc-Pro, one of the potyvirus-encoded proteinases approximately 50 kDa in size, is one of the well-known RSSs [31]. Hc-Pro was reported to counteract the defensive barrier by targeting multiple steps of the RNAi pathway. The Hc-Pro of *Tobacco etch virus* (TEV) hinders the assembly of siRNAs into RISC complexes by the direct binding to and sequestration of siRNAs [32]. Hc-Pro can also interfere with two key enzymes of the methionine cycle, as S-adenosyl-L-methionine synthase and S-adenosyl-L- homocysteine hydrolase, to inhibit the methylation of the vsiRNAs (virus siRNA) at the 3′ end [33], or directly interact with the HEN1 protein, which is a major methyltransferase in plant RNAi [34]. Hc-Pro has three domains: the N-terminal domain (1–100 amino acids) is involved in the functions of aphid transmission and genome replication; the central domain (101–299 aa) is related to RNA-silencing suppression, the viral particle yield and viral movement; and the C-terminal domain (300–459 aa) possesses proteolytic activity [35]. Many motifs and key residues were identified to play a critical role in these three domains, such as the KITC and PTK motifs for aphid transmission, IGN motif for genome amplification, CC/SC motif for long-distance movement and GYCY motif for cysteine protease activity [36,37,38,39]. The C-terminal region of Hc-Pro harbors a more conserved amino acid sequence than the N-terminal and the central domains [40]. However, most of the Hc-Pro functions are provided by the central domain, in which the F(Y)RNK box is a highly conserved motif located at amino acids 180–184 and participates in multiple functions including symptom severity, genome amplification, virus movement, RNA-silencing suppression and siRNA-binding activity [38,41]. There are significant differences in the RSS activity of Hc-Pro from different potyviruses. For example, the TEV and *Zucchini yellow mosaic virus* (ZYMV) Hc-Pro has a high RSS activity, while the *Potato virus Y* (PVY) and *Wheat streak mosaic virus* (WSMV) Hc-Pro shows weak RSS activity [38,42]. Hc-Pro may be stabilized by its adjacent P1 protein to produce effective RSS activity [43]. 

For the potyvirus Hc-Pro, the active form is dimers or multimers, which can be attributed to Hc-Pro self-interactions [44]. Size exclusion analysis suggests that the *Tobacco vein mottling virus* (TVMV) and *Turnip mosaic virus* (TuMV) Hc-Pro behaves as dimers or trimers [45]. Some reports demonstrated that only the N-terminal domain was involved in the Hc-Pro self-interactions of *Lettuce mosaic virus* (LMV) and PVY; however, other research identified interaction sites in both the N-terminal and the C-terminal domains in *Potato virus A* (PVA) Hc-Pro [45,46]. In other research, the N-terminus of the LMV Hc-Pro was reported to be nonessential for self-oligomerization, but the central and C-terminal regions were involved in protein interactions [47]. Yeast two-hybrid (Y2H) and bimolecular fluorescence complementation (BiFC) assays also illustrated that the central and C-terminal domains of the Hc-Pro participated in homodimerization [48]. Collectively, the self-interacting domains differ among the different potyviral Hc-Pros. 

In the present study, we constructed a functional full-length cDNA infectious clone of SMV-GFP using the yeast HR system for the first time. A tag-free SMV infectious clone was also obtained by Gibson assembly. Both infectious clones showed strong infection ability in *N. benthamiana*, soybean and several other legume species, and were confirmed to be seed transmitted in soybeans. Hc-Pro from SMV was shown to form oligomers, and Y2H and BiFC assays demonstrated that the specific region of oligomerization was located in the central and C-terminal domains. Twelve mutants were generated for key amino acids associated with RSS activity in Hc-Pro by site-directed mutation. The co-expression of mutant 2, which contains an Asn182Leu replacement in a FRNK motif, can cause attenuated symptoms and growth recovery in SMV-GFP infected plants, and it appears to enhance Hc-Pro self-interactions. 

## 2. Materials and Methods

### 2.1. Assembly of Full-Length cDNA Infectious Clones for SMV

SMV stocks were provided by Prof. Qingshan Chen from Northeast Agricultural University in China. A mild isolate of SMV-N1 was propagated in the Hefeng 25 soybean and was harvested and maintained at −80 °C as stocks. Total RNAs were isolated from virus-infected seedlings and reverse transcribed with the HiScript III 1st Strand cDNA Synthesis Kit (+gDNA wiper) according to the manufacturer’s instructions. To build infections clones of SMV, the SMV genome was split into four overlapping fragments: 1# (P1 + Hc-Pro, 2455 bp), 2# (P3 + 6K1 + CI + 6K2, 3282 bp), 3# (NIa-Vpg + NIa-Pro, 1323 bp) and #4 (NIb + CP, 2659 bp) by RT-PCR amplification using the primers listed in Appendix A. The binary vector pLXB (5# fragment) was linearized by PCR amplification using the PrimeSTAR^®^ GXL DNA Polymerase (Takara, Japan, Tokyo). The five fragments were gel-purified using the Thermo Scientific™ Gene JET™ GeL Extraction Kit, and Gibson assembly was performed using the NEBuilder HiFi DNA assembly master mix (New England Biolabs, United States). The ligate was transformed into the *E. coli* strain Trans2 Blue, and the positive colonies were confirmed by restriction endonuclease digestion, PCR amplification and Sanger sequencing. The correct assembled recombinant virus (pXLB-SMV, or SMV in brief) was transformed into *Agrobacterium tumefaciens* strain GV3101 for agroinfiltration. 

In order to generate the GFP-tagged SMV infectious clone by yeast HR, the *GFP* ORF sequence (792 bp) was amplified from pJL24 (a TMV-GFP infectious clone) [49] and was designed to be inserted in-frame between the *NIb* and *CP* genes of SMV. The SMV genome was divided into five overlapping fragments: A (P1 + Hc-Pro, 2446 bp), B (P3 + 6K1 + CI + 6K2, 3282 bp), C (NIa-Vpg + NIa-Pro, 1323 bp), D (NIb, 1608 bp) and F (CP, 1095 bp), respectively. The vector backbone pCB301–2µ-HDV (7838 bp) was linearized between the *CaMV* 35S promoter and the *tNOS* sequence by PCR amplification. All of the six PCR insert products and the linearized vector were designed to share a 20 bp homologous sequence at their borders. To assemble the plasmid with the yeast HR system, equivalent molar ratios of linearized vector, the five viral cDNA fragments and *GFP* fragment were co-transformed into yeast cells (Y2H Gold). Positive colonies were screened on tryptophan-minus media at 30 °C for at least 2–4 days, and the recombinant plasmids were isolated from the yeast cells. Positive colonies were confirmed by restriction endonuclease digestion and sequencing. The correct assembled recombinant virus (pCB301-SMV-GFP, or SMV-GFP in brief) was transformed into *Agrobacterium tumefaciens* strain GV3101 for agroinfiltration.

### 2.2. Virus Infection, Determination of Host Ranges, Detection of SMV by ELISA and Western Blots

*N. benthamiana* infection was performed according to Diao et al. [50], and soybean infection was performed according to Bao et al. [51]. *N. benthamiana* plants were leaf infiltrated with GV3101 containing SMV infectious clones at OD_600_ = 1.0 in infiltration buffer (1M MES, 1M MgCl_2_, 200 µM acetosyringone). The *N. benthamiana* leaves with leaf rolling and mosaic symptoms were collected at 5 dpi (days post infection) and used for mechanical inoculation assays on soybean and other legume species including the vegetable soybean, *Vigna unguiculata*, *Vigna angularis*, *Vigna radiata*, *Phaseolus* v1, *Phaseolus* v2, *Vicia faba*, *Astragalus sinicus*, *Medicago falcata*, *Glycine max* v1, *Pisum sativum*, *Phaseolus* v3 and *Lablab purpureus*. Five grams of leaf tissue was ground in 20 mL of 0.02 M potassium phosphate buffer (PH 7.0) with 10 mg/L silicon carbide. Seedlings at the three-leaf stage were rub inoculated with the above inoculum, and inoculation buffer without SMV was used as a mock treatment. The seedlings were grown in a greenhouse at 22–25 °C with a photoperiod of 14:10 (light/dark) and relative humidity of 70%. Symptom development was observed after infection at desired time points. Successful infection was also verified by the RT-PCR amplification of virus *CP* fragments or GFP mRNA via specific primers listed in Appendix A. SMV and SMV-GFP infections were detected by ELISA (mlBio ELISA kit) according to manufacturer’s instructions using virus-infected *N. benthamiana* leaf samples. The same samples were also used for detecting SMV CP protein expression by Western blotting using a CP-specific antibody. 

### 2.3. Recombinant Vector Constructions for Hc-Pro Mutants

The full-length ORF sequence of Hc-Pro was PCR amplified from SMV infectious clone DNA and cloned into the pMD19-T vector via TA cloning (Takara). Twelve RSS-activity-related mutants for Hc-Pro were obtained using this recombinant vector as a template with the Mut Express MultiS Fast Mutagenesis Kit (Vazyme, Nanjing, China) according to the manufacturer’s instructions. The twelve individual mutations are R181I, N182L, K183A, CDNQLD198-203AAAAAA, N200S, RKN240-243AAA, RE247-248AA, GS252-253DE, IGS251-253RAP, C344S, DE360-361AA and DH411-412AA, and the mutants are named as Hc-Prom1-m12. Wild-type Hc-Pro and the mutants were ligated into the binary vector pCambia-1300 with the ClonExpress^®^II One Step Cloning Kit (Vazyme, C11-01). All the constructs were confirmed by *Sal*I/*kpn*I restriction analysis and Sanger sequencing. 

### 2.4. Virus Inoculation and Agroinfiltration

The recombinant vectors and SMV infectious clones were transformed into *Agrobacterium tumefaciens* strain GV3101 by electroporation. *A. tumefaciens* was grown at 28 °C, centrifuged for collection, and diluted to OD_600_ = 1.0 or the indicated concentrations in infiltration buffer (1M MES, 1M MgCl_2_, 200 µM acetosyringone), and stored at room temperature for three hours. The inoculum was infiltrated into the abaxial side of full-grown leaves of *N. benthamiana* plants using a 1 mL syringe without a needle. For the mutant assay, the *A. tumefaciens* carrying the constructs of SMV-GFP and twelve Hc-Pro mutants (Hc-Prom1-m12) were mixed at a ratio of 1:1 for inoculation. GFP fluorescence was detected with a handheld long-wave (352 nm) UV lamp. 

### 2.5. Yeast Two-Hybrid (Y2H) and Bimolecular Fluorescence Complementation (BiFC) Assays

In order to construct plasmids for Y2H analysis, the full-length ORF sequences of wild-type Hc-Pro and mutant 2 (N182L) were PCR amplified and ligated into the pGADT7(AD) and pGBKT7(BD) vectors, respectively. To determine the homodimerization region for Hc-Pro, the central and C-terminal fragment (amino acid residues 287 to 457) of Hc-Pro were also ligated into the AD and BD vectors. The AD and BD recombinant vectors were co-transformed into the Y2H Gold yeast strain, and the transformants grown on SD/-Try/-Leu medium were confirmed by colony PCR. The yeast cells were transferred to SD/-Try/-Leu/-His/X-α-Gal media for 3–7 days at 30 °C. The fully grown colonies as well as blue-colored colonies were photographed. All experiments were repeated at least three times.

For the BiFC assays, the wild-type, mutant 2 (N182L) and truncated Hc-Pro sequences were cloned into the pDEST-nYFP and pDEST-cYFP vectors, fused with the yellow fluorescent protein (YFP) at the N-terminal region (nYFP) or the C-terminal region (cYFP), by Gateway cloning. Recombinant plasmids were transformed into *Agrobacterium tumefaciens* strain GV3101 and infiltrated into *N. benthamiana* leaves. Infected leaves were analyzed 72 h after infiltration, and the YFP and DAPI fluorescence were detected using a confocal laser-scanning microscope.

### 2.6. RNA Extraction and PCR Analysis

The experimental procedures followed the literature [52]. Briefly, total RNA was isolated from the leaves or seeds by the TRIzol reagent (CWBio, Beijing, China) extraction method following the manufacturer’s instructions. The quality and concentration of the total RNA were analyzed with a NanoDrop2000 spectrophotometer and by resolution via 1% agarose electrophoresis. Genomic DNA contamination was removed by DNase treatment. One microgram of mRNA was reverse transcribed with the HiScript III 1st Strand cDNA Synthesis Kit (+gDNA wiper) according to the manufacturer’s instructions. The Fast SYBR Green Master Mix (TransGen Biotech, Beijing, China) was used, and the qTOWER 2.2 (Analytik Jena, Germany) equipment was used for the RT-qPCR experiments. The mRNA levels of *N. benthamiana* or soybean *β-actin* were used as internal controls. The ratio of mRNA abundance between the treatment and mock samples was calculated by the 2^−ΔΔCT^ method for each gene. Three biological replicates were performed, and a T-test was performed between the treatment and the control mock treatment groups. 

## 3. Results

### 3.1. One-Step Assembly of SMV cDNA Infectious Clone and Its Infectivity

pLXB is a mini binary vector suitable for type II endonuclease digestion and overlapping-based assembly [53]. Because of its large genome size, the SMV genome (N1 strain, GenBank Accession No. D00507.2) was divided into four fragments for PCR amplification. The four fragments were assembled sequentially into the linearized vector (pLXB, 4198 bp) in a one-step reaction, as they shared homologous sequences at each boundary (Appendix A). When the ligate was transformed into *E. coli* and recombinant colonies were recovered, they showed the expected size (13,917 bp) upon *Sal*Ι digestion (Figure 1A), and sequencing analysis further confirmed the successful assembly of the full-length SMV infectious clone (pLXB-SMV, SMV). 

To evaluate its infectivity, the recombinant pLXB-SMV was transformed into *A. tumefaciens* GV3101 and agroinfiltrated on *N. benthamiana* seedlings followed by being rub-inoculated on susceptible soybeans. The RT-PCR results showed that SMV could be detected in the systemic leaves of *N. benthamiana* and soybean plants (Figure 1B). In *N. benthamiana*, pLXB-SMV-inoculated plants began to show leaf mottling and curling symptoms at 6 days post infiltration (dpi), and systemic infection occurred in all of infected plants in the newly grown leaves at 10 dpi (Figure 1C). The disease onset in the soybean plants rub-inoculated with the pLXB-SMV clone was generally delayed by 4 days compared with that in *N. benthamiana,* while it was still characterized by leaf yellowing, mosaic and curling (Figure 1D). The results showed the pathogenicity and systemic spread of SMV derived from the infectious clone in *N. benthamiana* and soybeans. 

### 3.2. Construction of GFP-Tagged SMV Infectious Clone and Infection Assays

To visualize SMV infection for easier detection, we used the yeast–agrobacterium shuttle vector (pCB301-2µ-HDV) to construct the SMV-GFP infectious clone via yeast homologous recombination. To this end, the SMV genome was amplified into five fragments with 20 nt homologous termini. Meanwhile, the ORF sequence for GFP (green fluorescent protein) was cloned from the TMV-GFP vector (pJL24) and inserted in-frame between the *NI*b and *CP* genes (Appendix A). To maintain the biological properties of the parental SMV isolate, an artificial viral proteinase cleavage site (TYEVHHQ) was added at both the N’ and C’ sides of the GFP sequence, allowing the proteolytic maturation of GFP from viral polyproteins. Recombinant plasmids were confirmed with *Kpn*Ι digestion (Appendix A) and Sanger sequencing. The recombinant SMV-GFP was transformed into *A. tumefaciens* GV3101, agroinfiltrated to *N. benthamiana* seedlings and subsequently rub-inoculated on susceptible soybean plants. In *N. benthamiana*, GFP fluorescence began to appear on infected leaves at 4 dpi, spread to the uninfected areas at 7 dpi, and was detected in all of systemic leaves at 11 dpi. Accordingly, *GFP* mRNA expression was also confirmed by a RT-PCR experiment (Appendix A). 

Meanwhile, SMV-GFP-infected *N. benthamiana* showed typical SMV phenotypes such as curling, mottling and mosaic (Figure 2A). Similar symptoms were also observed, while GFP fluorescence was very dim in plants of the susceptible soybean varieties Williams and Hefeng 25 (Figure 2B). SMV-GFP was also detected by RT-PCR in the systemic leaves of both *N. benthamiana* and soybeans (Figure 2C). Taken together, the above results show that the SMV-GFP infectious clone has strong infectivity and can stably express GFP. Based on the above results, we further examined virus protein expression and virus accumulation in plants. SMV CP expression was detected by Western blotting (Figure 3A), and virus accumulation was also confirmed by ELISA (Figure 3B) for both the SMV and SMV-GFP infectious clones.

### 3.3. Seed Transmission of Infectious Clone-Derived SMV in Soybeans

SMV was reported to be seed transmitted in soybeans. In order to investigate the seed transmission properties of SMV derived from our infectious clones, seeds were collected from the SMV-susceptible Williams and Hefeng 55 soybean cultivars after infections. The seeds showed mottling symptoms in both varieties after SMV infections resulting from both infectious clones (Figure 4A), and seed weight was significantly decreased (Figure 4B). Seed mottling was reported to be associated with the increases in chalcone synthase (CHS) expression. Consistent with this, the *CHS* mRNA level was significantly increased in the mottled seeds compared with in the control healthy seeds (Figure 4C). When brown-mottled seeds from different soybean varieties were planted in pots, the leaves in newly grown seedlings exhibited the typical SMV-induced phenotypes such as curling and yellowing (Figure 4D). Furthermore, the brown-mottled seeds showed a significant reduction in germination rates compared with the healthy seeds (Figure 4E). The RT-PCR results also indicated that SMV was present in progeny seedlings (Figure 4F). Those results confirmed the transmission of the SMV derived from our infectious clones to progeny plants via the seeds.

### 3.4. Host Ranges of SMV Derived from Infectious Clones

In order to detect the broad-spectrum infectivity of the SMV derived from infectious clones, thirteen plant species of legumes were infected via rub inoculation with both pLXB-SMV and SMV-GFP. Plants of five out of the thirteen species, including *Phaseolus vulgaris* v1, *Vigna angularis*, *Vigna unguiculata*, *Phaseolus vulgaris* v2 and *Glycine max* v1 (vegetable soybean), showed SMV symptoms ranging from typical mosaic to local necrotic lesions and leaf malformation (Figure 5A, Table 1). 

RT-PCR analysis also confirmed the presence of SMV in the leaves of these plant species (Figure 5B). By contrast, eight other legumes including *Vigna radiata*, *Vicia faba*, *Astragalus sinicus*, *Medicago falcata*, *Glycine max* v2 (black bean), *Pisum sativum*, *Phaseolus vulgaris* v3 and *Lablab purpureus* showed no viral symptoms. The results demonstrated the infectivity and utility of the infectious clones in some legume species, although a limited host range was observed.

In all of the above plant species, SMV induced more severe virus symptoms than SMV-GFP. Therefore, we quantified the virus titers in these plant species via RT-qPCR experiments. Slightly higher virus accumulation was observed for SMV than SMV-GFP, which might partly explain the more serious virus symptoms in SMV than SMV-GFP (Figure 5C). 

### 3.5. The Effects of Mutant Hc-Pro on SMV Infectivity and Symptom Development

After the successful establishment of infectious clones, we used them, especially SMV-GFP, to characterize the functions of Hc-Pro, a silencing suppressor of SMV. In order to verify the self-interaction of Hc-Pro and identify the peptide region that was required for Hc-Pro self-interaction, a series of recombinant plasmids for Y2H and BiFC assays were constructed. The rationale of the Y2H experiment is shown in Appendix A. The full-length Hc-Pro showed self-interactions, while the truncated Hc-Pro^170^ displayed both stronger self-interactions and inter-molecular interactions with the full-length Hc-Pro. The results demonstrated that the interaction sites of SMV-Hc-Pro are mainly located in the central and C-terminal domains (Figure 6A). 

To determine Hc-Pro in planta interactions, we performed BiFC assays in *N. benthamiana*. The rationale of the BiFC experiment is shown in Appendix A, and the detection of recombinant BiFC vectors is displayed in Appendix A. When the full-length Hc-Pro or Hc-Pro^170^ was fused to the N- or C-terminal domain of a split yellow fluorescent protein (sYFP) and co-inoculated into *N. benthamiana* leaves, strong YFP fluorescence could be detected in the cytoplasm and, especially, in the nucleus (Figure 6B). 

Twelve mutations for Hc-Pro RSS activity-related residues were generated, including R181I, N182L, K183A, CDNQLD198-203AAAAAA, N200S, RKN240-243AAA, RE247-248AA, IGS251-253RAP, GS252-253DE, C344S, DE360-361AA and DH411-412AA, abbreviated as m1–m12 (Figure 7A). The role of mutant Hc-Pros was determined by the co-infection of the SMV-GFP (containing a native Hc-Pro) with the binary vectors over-expressing native or mutant Hc-Pro in *N. benthamiana*. The positive control (P19) strongly increased the local GFP fluorescence intensity in agroinfiltrated *N. benthamiana* leaves at 5 dpi. However, the GFP fluorescence intensity was not improved by Hc-Pro. Interestingly, Hc-Prom2 resulted in the disappearance of GFP fluorescence on all of inoculated plants among the twelve Hc-Pro mutants (Figure 7B–D). The point mutation N182L in Hc-Prom2 was located at the FRNK motif, which has been reported to be responsible for RSS activity. The possible reason for the disappearance of GFP upon Hc-Prom2 expression is that it competitively interacts with native Hc-Pro from SMV-GFP, and this interaction abolishes native Hc-Pro activity. This assumption was validated by Y2H experiments, in which the Hc-Prom2 self-interactions and interaction with Hc-Pro were stronger than that of Hc-Pro itself (Figure 7E). On the other hand, Hc-Prom2 showed notable alleviative effects on SMV-infected plant growth and symptom development. When Hc-Prom2 was co-expressed with SMV-GFP, Hc-Prom2 abolished SMV symptoms, resulting in the recovery of plant growth, decreased GFP expression and the attenuation of SMV-typical phenotypes such as curling and mottling at 25 dpi (Figure 8).

## 4. Discussion

Host–virus interaction is one of the important topics in the field of plant virology. In this regard, virus infectious clones are powerful reverse genetics tools and become a limiting factor for some virus studies [54]. The construction of full-length cDNA infectious clones for potyviruses was reported to be technically challenging, owing to their large genome size and strong toxic effect on the virus components on *E. coli* during transformation [55,56,57]. Several strategies such as utilizing low-copy vectors, in vitro ligation [58,59,60], intron insertion [24,55,56,61] and yeast homologous recombination (HR) [62,63,64] have been used to overcome these problems. In recent years, several simple and effective in vitro assembly methods, such as multiS one-step clones, in-fusion cloning and Gibson assembly, have also been used to construct infectious clones [24,65,66,67]. Yeast HR-based cloning methods utilize yeast to assemble multiple DNA fragments via the 20–30 bp homology of ends between the two adjacent fragments. The yeast HR-based cloning method is simple and efficient, and infectious clones for several animal and plant RNA viruses have been successfully constructed by this method [22,63]. In this study, we first used Gibson assembly to construct an SMV infectious clone and then successfully assembled the GFP-tagged SMV infectious clone by yeast HR. Both of the infectious clones are suitable for infection via agroinfiltration, and the resulting viruses are seed transmitted. Typical SMV symptoms, such as leaf curling, mottling and necrosis, can be detected on the systemic leaves of the model plant *N. benthamiana* and the natural host soybean. The SMV-GFP infectious clone has strong infectivity and efficiently expresses the GFP. Although several SMV infectious clones have been developed by different approaches [68,69,70], the present study is the first report for SMV-GFP. Virus symptom development is influenced by many factors and is sometimes difficult to be observed, or plants can sometimes remain symptomless. Therefore, the visually detectable SMV-GFP construct developed here is a powerful tool for tracking SMV infection and further studying the host–virus interactions.

SMV can be transmitted through seeds and causes seed mottle on the epidermis [14,15]. Previous studies showed that the extent of seed pigmentation was dependent on host genotypes, SMV isolates, the interactions between the soybean cultivar and SMV isolate [71,72,73,74], and the soybean growth stage at the time of infection [75]. SMV-induced seed pigmentation was associated with the suppression of the tissue-specific silencing of *CHS* gene expression [76]. In the present study, the seed transmissibility of the viruses derived from the infectious clones—pLXB-SMV and SMV-GFP—was confirmed by seed mottling and even typical SMV symptoms in their progeny plants. Seed-borne viruses are agriculturally important because the virus may persist in the seeds for a period of time and may spread the disease to new areas. Several seed-transmitted plant viruses such as *Pea seed-borne mosaic virus* (PSbMV) in *Pisum sativum*, *Yellow mosaic virus* (YMV) in *Vigna mungo*, *Beet curly top virus* (BCTV) in *Petunia hybrida*, *Cucumber mosaic virus* (CMV) in *Capsicum annuum*, *Sweet potato leaf curl virus* (SPLCV) in *Ipomoea batatas*, and *Zucchini yellow mosaic virus* (ZYMV) in *Cucurbita pepo* ssp. *Texana* were reported [71]. Besides SMV, *Soybean vein necrosis virus* (SVNV) and *oTmato yellow leaf curl virus* (TYLCV) were also reported to be seed transmitted in soybeans [71,76,77,78,79]. However, there were limited reports on the seed transmissibility of viruses derived from infectious clones of plant viruses. Therefore, the SMV infectious clones will be valuable tools for studying seed transmission properties and mechanisms for SMV and other related virus species.

SMV’s host range is reported to be relatively narrow, and the natural host range of SMV is mainly restricted to legumes such as *G. max* (cultivated soybean) and *G. soja* (wild soybean). Apart from soybean, most strains only systematically infect several leguminous plants such as *Vigna angularis*, *Phaseolus vulgaris*, *Vigna radiata*, *Pisum sativum*, *Vicia faba* and *Astragalus sinicus*. In addition, some non-leguminous plants including *Chenopodium album*, *Senna occidentalis* and *Pinellia ternate*, *Passiflora* spp. could also be infected by SMV through mechanical or natural inoculation [71,80,81,82,83,84,85,86]. In this study, we found that five out of the 13 natural and experimental host legume plants, including *Phaseolus vulgaris* v1, *Vigna angularis*, *Vigna unguiculata*, *Phaseolus vulgaris* v2 and *Glycine max* v1 (vegetable soybean), can be systemically infected by SMV derived from our infectious clones. They showed similar symptoms to cultivated soybeans on the leaves, including mosaic, rugosity and leaf curling. By contrast, eight other legumes including *Vigna radiata*, *Vicia faba*, *Astragalus sinicus*, *Medicago falcata*, *Glycine max* v2 (black bean), *Pisum sativum*, *Phaseolus vulgaris* v3 and *Lablab purpureus* showed no viral symptoms. The results demonstrated the selective infectivity and usefulness of the infectious clones in some legume species, although a limited host range was observed.

Hc-Pro is one of the most widely studied multifunctional proteins in potyviruses [47,87], which is mainly involved in aphid transmission [88], the suppression of the RNA silencing [31], virus movement, symptomatic expression and other functions [89]. The self-interacting capacity of Hc-Pro has been shown in many potyviruses, such as LMV, PVY and TVMV [44,90]. The oligomerization of Hc-Pro seems to be indispensable for its biological functions. Previous studies have shown that the conserved FRNK motif of *Papaya leaf distortion mosaic virus* (PLDMV) and ZYMV R→I mutation significantly reduced the viral symptoms, and the same mutation in Hc-Pro from PVY abolished its RSS activity [38]. In addition, the mutation in the CDNQLD motif downstream of FRNK could negatively affect the RSS activity of Hc-Pro in *Tobacco vein banding mosaic virus* (TVBMV) [91]. In the present study, the major region responsible for homodimerization was confirmed in the central 13-amino-acid region and whole 157-amino-acid C-terminal region of Hc-Pro in the case of SMV. Furthermore, a N182L mutation in the FRNK motif could alleviate SMV symptoms, which seemed to be attributable to the increased interaction of this mutant with native Hc-Pro and its competitive actions. Therefore, the conserved motif of potyvirus Hc-Pro FRNK may be involved in its oligomerization, thus affecting its pathogenicity and symptom severity. This mutant could also be useful for the future control of SMV, as the overexpression of this mutant might endow immunity against invading SMV.

## 5. Conclusions

Soybean mosaic virus (SMV) is a severe soybean pathogen posing threats to soybean cultivation, and the molecular mechanisms underlying SMV–soybean interactions are poorly known. In view of this, we developed two infectious clones for SMV—in particular, SMV-GFP—and we investigated their infectivity in plant hosts and their utility in functional gene studies. The infectivity of SMV derived from both infectious clones for the natural host soybean and several other legume species makes them useful tools for studying host–SMV interactions. Especially, the infectivity and efficient GFP expression of the SMV-GFP infectious clone might facilitate functional studies of both virus and host genes using the *N. benthamiana* transient expression system. Furthermore, we identified a key amino acid for SMV infectivity and symptom development in the FRNK motif of the silencing suppressor Hc-Pro. Taken together, our data will help to expedite the understanding of molecular mechanisms for soybean–SMV virus interactions. 

## Figures and Tables

**Figure 1 viruses-12-00886-f001:**
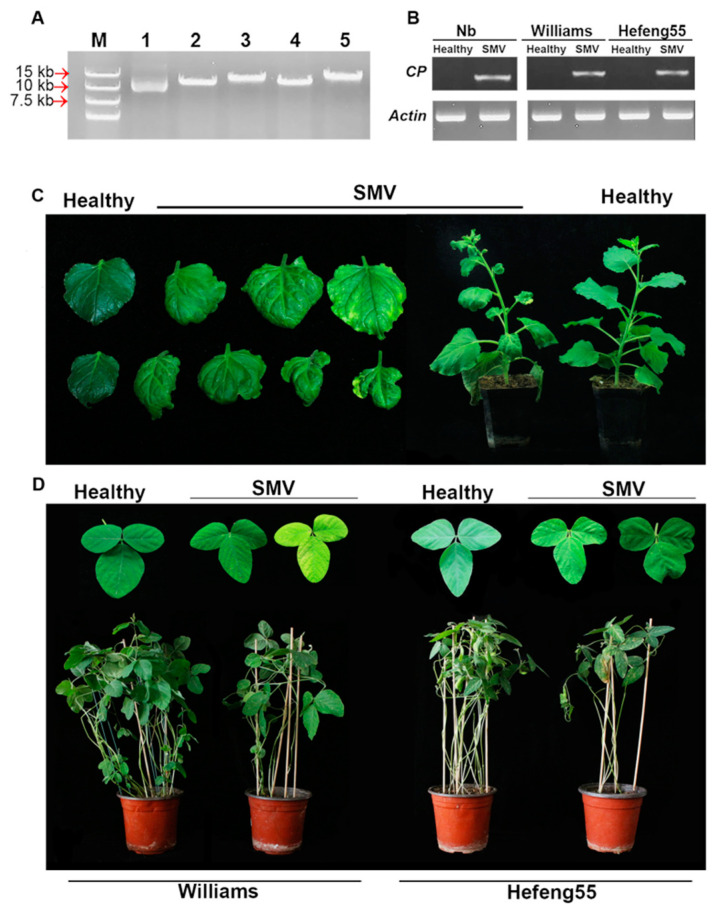
Infectivity of *Soybean mosaic virus* (SMV) derived from the pLXB-SMV infectious clone in *N. benthamiana* and soybean. (**A**) Confirmation of the pLXB-SMV infectious clone by restriction endonuclease digestion. M: DL15000 marker. Lane 1: plasmid control without enzyme digestion. Lanes 2–5: individual recombinant plasmids digested with *Sal*I. (**B**) RT-PCR analysis of SMV via coat protein region (CP)-specific primers in *N. benthamiana* and soybean leaves at 12 days post infection (dpi). The internal reference gene *actin* was used as a control. (**C**) Symptoms of pLXB-SMV-infected *N. benthamiana* plants and representative leaves. (**D**) Symptoms of pLXB-SMV infected soybean plants and representative leaves.

**Figure 2 viruses-12-00886-f002:**
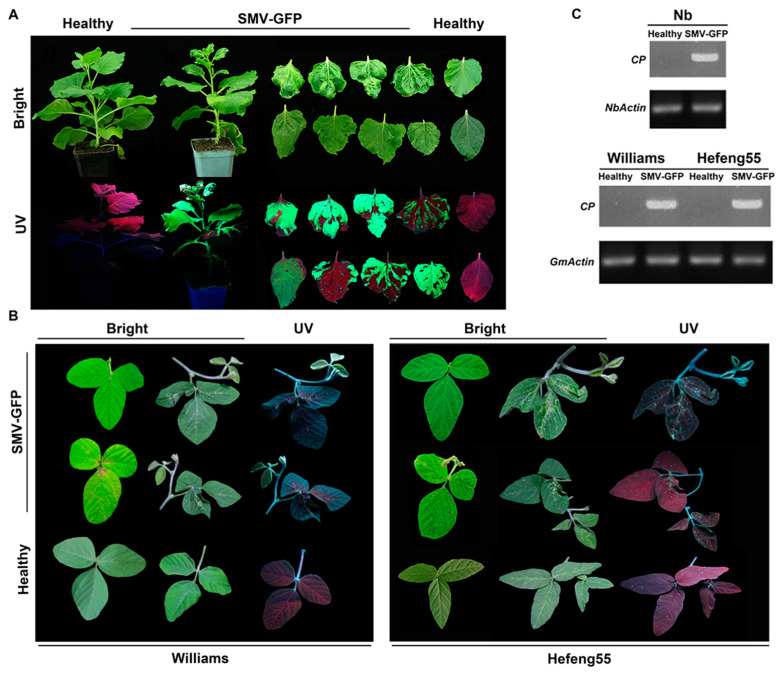
Infectivity of SMV-GFP infectious clone in *N. benthamiana* and of SMV-GFP in soybeans. (**A**) Symptoms of SMV-GFP-infected *N. benthamiana* plants and representative leaves. Bright, natural light. UV, 352 nm. (**B**) Symptoms of SMV-GFP infected soybean plants and representative leaves. (**C**) RT-PCR analysis of SMV via coat protein (CP) region-specific primers in *N. benthamiana* and soybean leaves at 12 dpi. The internal reference gene *actin* was used as a control.

**Figure 3 viruses-12-00886-f003:**
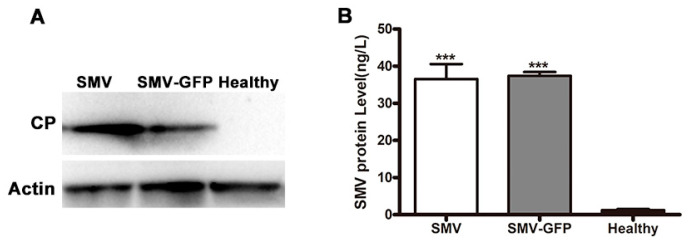
Detection of SMV protein and virus accumulation for two infectious clones in plants. *N. benthamiana* plants were inoculated with SMV or SMV-GFP infectious clones, and the infected leaves were collected 5 dpi for total protein isolation. (**A**) Detection of SMV CP protein via Western blotting. The internal reference protein actin was used as a control. CP, 29 kDa; actin, 48 kDa. (**B**) Detection of SMV accumulation via ELISA. T-test was performed between healthy and SMV or SMV-GFP infected plants. Triple asterisks, *p* < 0.001.

**Figure 4 viruses-12-00886-f004:**
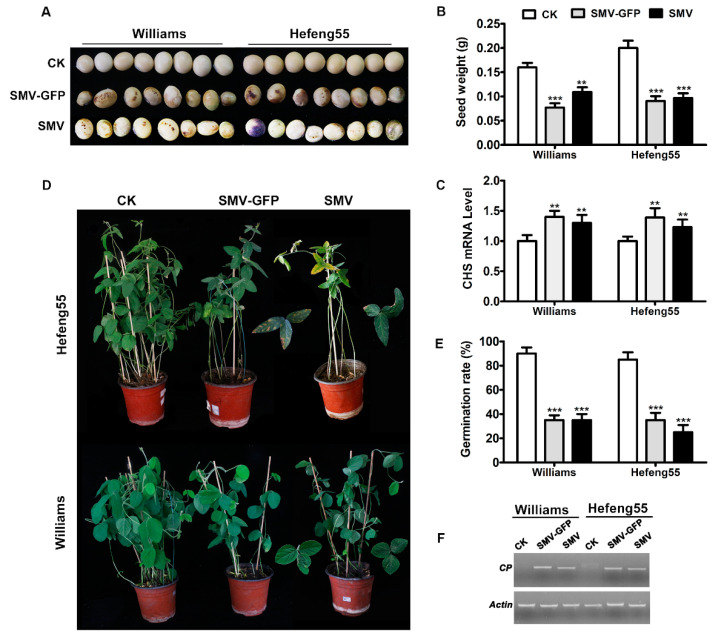
Seed transmission of SMV derived from infectious clones. (**A**) Visual appearance of seed mottling upon infection with SMV derived from infectious clones. (**B**) Weight of the seeds showing mottling upon infection with SMV from infectious clones. T-test was performed between treatment and control group. Double asterisks, *p* < 0.01, Triple asterisks, *p* < 0.001. (**C**) *CHS* mRNA level determined by RT-qPCR assay in mottling seeds. Soybean *actin* gene expression was used as an internal control. (**D**) Phenotype of SMV-infected progeny seedlings. The mottling seeds were germinated in soil pots, and the phenotype was observed at 35 days. (**E**) Germination assay of SMV-infected seeds. The mottled seeds were germinated in soil pots, and the germination rate was calculated at 20 days. (**F**) SMV-specific RT-PCR analysis of progeny plants from SMV seed transmission. The internal reference gene *actin* was used as a control. The average number and SE value in each treatment group are shown. Statistical differences between SMV-infected and the control group were evaluated. Double asterisks indicate a significant difference at *p* < 0.01, and triple asterisks indicate a significant difference at *p* < 0.001.

**Figure 5 viruses-12-00886-f005:**
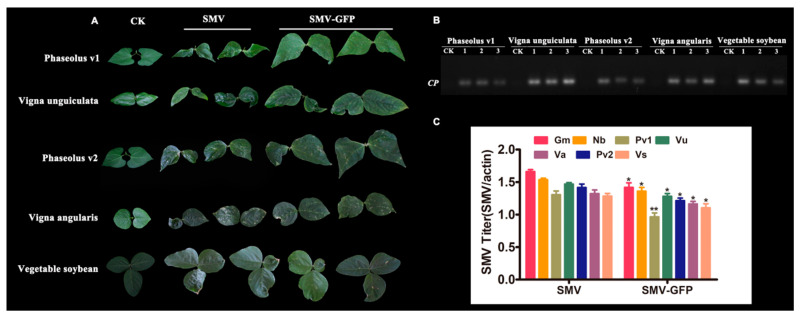
Infectivity of SMV from infectious clones in legume species. (**A**) Representative leaves with virus symptoms such as leaf curling, mosaic and local necrotic lesions were photographed. (**B**) RT-PCR analysis of SMV via coat protein (CP) region-specific primers is shown. CK, mock treatment; Line 1, SMV; Line 2, SMV-GFP; Line 3, positive control with the plasmid as template. (**C**) Detection of SMV titers in different plant species via RT-qPCR using *CP*-specific primers. Gm, *Glycine max*; Nb, *Nicotiana benthamiana*; Pv1, *Phaseolus* v1; Vu, *Vigna unguiculata*; Pv2, *Phaseolus* v2; Va, *Vigna angularis*; Vs, Vegetable soybean. Single asterisk, *p* < 0.05, Double asterisks, *p* < 0.01.

**Figure 6 viruses-12-00886-f006:**
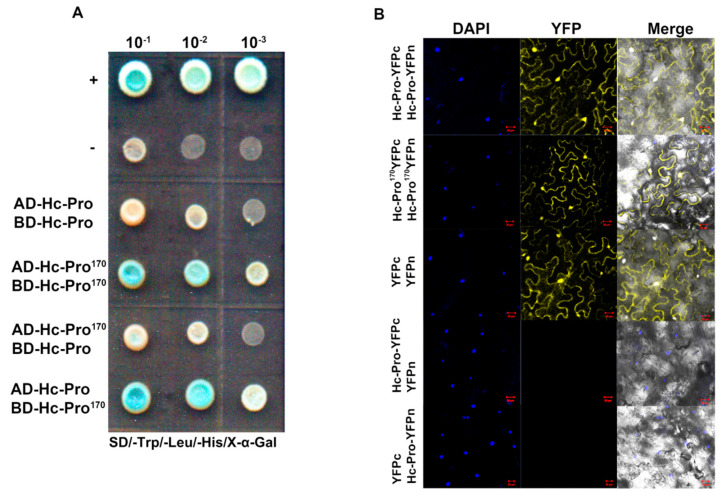
Yeast two-hybrid (Y2H) and bimolecular fluorescence complementation (BiFC) assays for Hc-Pro self-interaction. (**A**) Y2H assay. The yeast cells containing indicated constructs were grown on selective medium lacking leucine, tryptophan and histidine (X-α-Gal). The yeast concentration gradients were 10^−1^, 10^−2^ and 10^−3^, respectively. pGBKT7-p53/pGADT7-RecT (positive control) is symbolized as “+”, and pGBKT7-p53/pGADT7-lamin (negative control) is shown as “–”. (**B**) BiFC assay of Hc-Pro interactions in *N. benthamiana*. The full length Hc-Pro and truncated Hc-Pro^170^ were transiently expressed by co-infiltration on *N. benthamiana* leaves. sYFPn plus Hc-Pro-sYFPc or Hc-Pro-sYFPn plus sYFPc was used as the negative control. At 3 dpi, YFP fluorescence imaging was performed by confocal laser-scanning microscopy (488 nm). DAPI staining was performed to show the nuclei.

**Figure 7 viruses-12-00886-f007:**
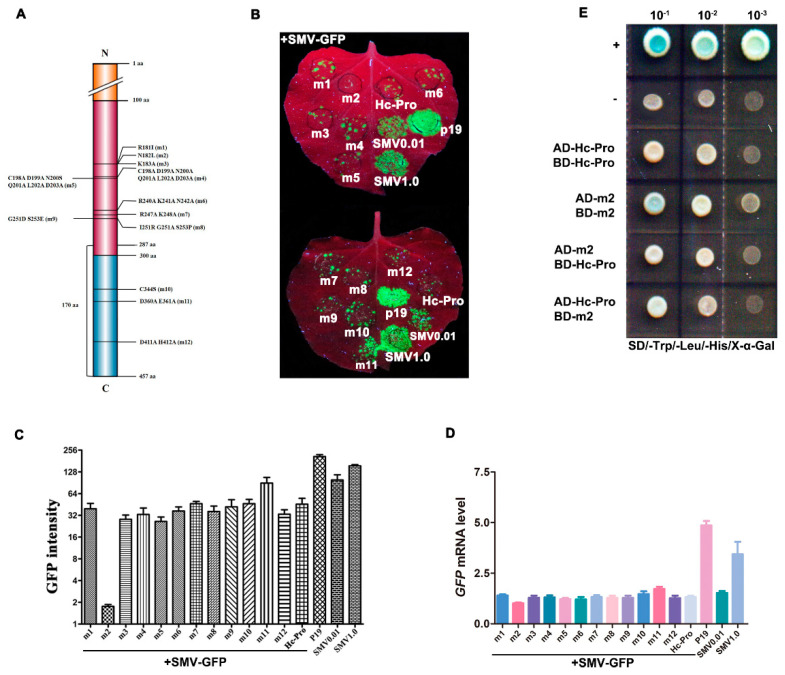
Functional analysis of mutant Hc-Pros. (**A**) Schematic diagram of mutation sites on Hc-Pro protein. Different colors indicate three different domains. (**B**) Effects of twelve Hc-Pro mutants on SMV-GFP infection. Each Hc-Pro mutant construct was co-infiltrated with SMV-GFP, and GFP fluorescence was visualized at 5 dpi under a UV lamp. P19 served as a positive control, and empty vector served as a negative control. (**C**) The GFP fluorescence in the infected area of 12 mutants in (B) was quantified with the Gel-Pro analysis software. (**D**) RT-qPCR detection of GFP mRNA level. **(E)** Detection of interactions between Hc-Pro and Hc-Prom2 by Y2H assays. The yeast cells containing indicated constructs were grown on selective medium lacking leucine, tryptophan and histidine (X-α-Gal). The yeast concentration gradients were 10^−1^, 10^−2^ and 10^−3^, respectively. pGBKT7-p53/pGADT7-RecT (positive control) is symbolized as “+”, and pGBKT7-p53/pGADT7-lamin (negative control) is shown as “–”.

**Figure 8 viruses-12-00886-f008:**
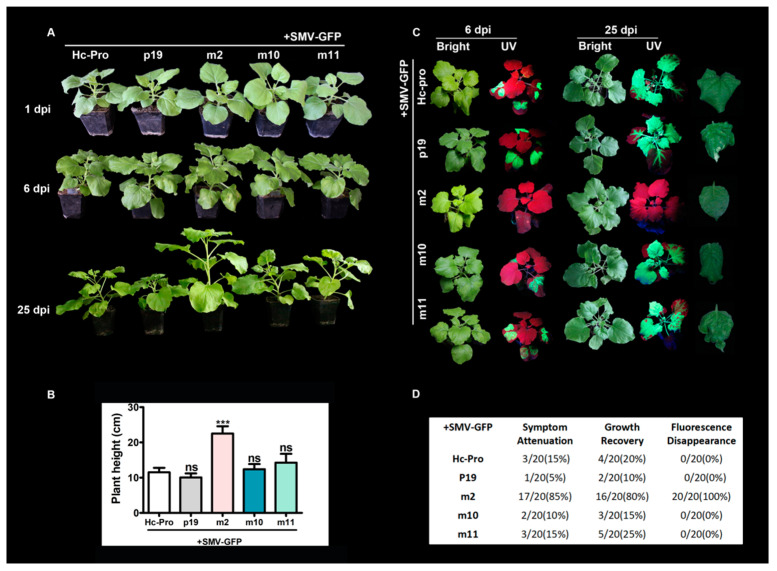
Effects of co-expression of Hc-Pro mutants on SMV-GFP infection. (**A**) Plant phenotypes at different days upon co-expression of Hc-Pro mutants and SMV-GFP infection. (**B**) Plant height. T-test was performed between treatment and control group. Triple asterisks, *p* < 0.001. ns, non-significant. (**C**) GFP fluorescence visualized under UV light. Representative leaves under bright light are shown on the right panel. (**D**) Symptom statistics.

**Table 1 viruses-12-00886-t001:** Symptoms in legume species upon infection with SMV infectious clones.

Host	Symptoms Caused by SMV	Symptoms Caused by SMV-GFP
*Vegetable soybean*	M, SN, LC	NLL, SN, LC
*Vigna unguiculata*	SN, LC	M, SN, LC
*Vigna angularis*	M, NLL, LC, SN	M, NLL, LC, SN,
*Vigna radiata*	N	N
*Phaseolus v1*	M, LC, SN	M, NLL, LC
*Phaseolus v2*	M, LC, SN	M, NLL, LC, SN
*Vicia faba*	N	N
Astragalus sinicus	N	N
*Medicago falcata*	N	N
*Glycine max v1*	N	N
*Pisum sativum*	N	N
*Phaseolus v3*	N	N
*Lablab purpureus*	N	N

M = mosaic; NLL = necrotic local lesion; SN = systemic necrosis; LC = leaf curling; N = no symptom.

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
