# Peer review of "Synthesis of Full-Length cDNA Infectious Clones of Soybean Mosaic Virus and Functional Identification of a Key Amino Acid in the Silencing Suppressor Hc-Pro"

_viruses, 2020, doi:10.3390/v12080886_

Round 1

Reviewer 1 Report

In the manuscript authors presented the utilization of Gibson assembly and yeast homologous recombination to construct infectious clones of SMV with and without GFP. Authors also verified if the virus HC-Pro and its mutants can self-interact and if they showed silencing suppressor activities, when overexpressed together with SMV-GFP.

While this method is going to be useful for other researchers, and authors provided a good set of experiments to verify the pathogenicity of the constructs in different hosts and in seed, few questions remain to be elucidated.

L39: has the function of SMV proteins being verified experimentally? If not, please add that these are putative proteins, otherwise report the references of the work showing the experimentally proven proteins’ function.

Is the infectious clone expressing all proteins correctly? Are virions formed and can they be observed in infected plants? Can a western blot with specific antibodies or an ELISA used to verify protein expression and virion formation? Even by RT-PCR, the only verified transcript was the one of partial CP RNA, I do not think this is enough to show that the infectious clone functions like the wt virus.

The same questions pertain to the infectious clone carrying the GFP fusion.

It is not clear to me to what the expressed GFP would be fused to and how this fusion was verified in planta. This should be shown and demonstrated.

  1. 149: add reference for pJL24 plasmid

L191: add more details or a reference to the agrobacterium growth and infiltration; for instance, what was the ‘appropriate’ OD600? Where the Agrobacterium suspensions stored for few hours before infiltration? Etc.

L 194 what is the 35S-Hc-Prom1-m12?

How was the success of the plasmid construction and protein expression (if they were full length for instance) of the BiFC assay verified?

Figures showing the BiFC assay in 5B are of poor quality. Only two cells show nuclei by DAPI, and the size of the nuclei seem overenlarged, suggesting that either the DAPI staining formed blobs or that the nuclei have been altered by the experimental conditions. Can the author provide better and more clear images, and can they address these concerns? Are there positive and negative controls included and indicated?

Figure 1c and 1, and 3d: pictures are dark and it is hard to see the symptoms on the infected leaves, provide better images and a close up of parts of the laminar blade showing symptoms.

Figure 2b: is the healthy leaf on the bottom middle switched with the bottom right?

Figure 6: why is the wt HC-Pro control not showing any silencing suppressor activity but some of its mutants are?

Author Response

Editor’s comments:

Manuscript ID: viruses-867567

Title: Synthesis of full-length cDNA infectious clones of Soybean mosaic virus and functional identification of a key amino acid in silencing suppressor Hc-Pro

Please revise the manuscript according to the reviewers' comments and upload the revised file within 10 days. Use the version of your manuscript found at the above link for your revisions, as the editorial office may have made formatting changes to your original submission. Any revisions should be clearly highlighted, for example using the "Track Changes" function in Microsoft Word, so that changes are easily visible to the editors and reviewers. Please provide a cover letter to explain point-by-point the details of the revisions in the manuscript and your responses to the reviewers' comments. Please include in your rebuttal if you found it impossible to address certain comments. The revised version will be inspected by the editors and reviewers. Please detail the revisions that have been made, citing the line number and exact change, so that the editor can check the changes expeditiously. Simple statements like ‘done’ or ‘revised as requested’ will not be accepted unless the change is simply a typographical error.

Responses to editor: Overall, the editor’s and reviewer’s comments we received in last submission are very insightful and helpful. We really appreciate the comments and thoroughly revised the manuscript based on their comments.

We made substantial improvements on the manuscript (Manuscript ID: viruses-867567) by the following. 1)We detailed the methods and data. 2) We added some new data. 3) We revised the Figures. 4) We revised the writings. Followings are the reply to per comment.

Please note that we showed the line numbers to clearly tell where some revisions were made, while we realized that sometimes this line number changed a little (by one or two numbers) when creating the merged file.

Reviewer #1 (Comments for the Author):

In the manuscript authors presented the utilization of Gibson assembly and yeast homologous recombination to construct infectious clones of SMV with and without GFP. Authors also verified if the virus HC-Pro and its mutants can self-interact and if they showed silencing suppressor activities, when overexpressed together with SMV-GFP.

While this method is going to be useful for other researchers, and authors provided a good set of experiments to verify the pathogenicity of the constructs in different hosts and in seed, few questions remain to be elucidated.

Q1:  L39: has the function of SMV proteins being verified experimentally? If not, please add that these are putative proteins, otherwise report the references of the work showing the experimentally proven proteins’ function.

Reply: We would say that the 11 component proteins of SMV have been identified in literatures while some of their functions were still not clear, so we corrected them as putative proteins, and here we cited references 4 and 5 on lines 41-45.

Q2:  Is the infectious clone expressing all proteins correctly? Are virions formed and can they be observed in infected plants? Can a western blot with specific antibodies or an ELISA used to verify protein expression and virion formation? Even by RT-PCR, the only verified transcript was the one of partial CP RNA, I do not think this is enough to show that the infectious clone functions like the wt virus.

The same questions pertain to the infectious clone carrying the GFP fusion.

Reply: We did two more experiments to detect the viruses (both SMV and SMV-GFP) in N. benthamiana. First, we did CP specific Western blot and the data were presented as Fig.3A. Second, we did ELISA to detect viruses and the results were shown in Fig. 3B.

Q3:  It is not clear to me to what the expressed GFP would be fused to and how this fusion was verified in planta. This should be shown and demonstrated.

Reply: SMV has a monocistronic genome and its proteins are first expressed (transcribed and translated together) as a polyprotein and cleaved by proteinases to produce mature proteins. We expressed GFP between NIb and CP proteins in frame. We clarified this in Line 151-152 and line 273-277. To show this clearly, we detect GFP mRNA expression with gene specific primers, and the results were provided in Supp. Fig. 2C.

Q4:  L149: add reference for pJL24 plasmid

Reply: Done on Line 151 by referencing John Lindbo et al. (ref 50).

Q5:  L191: add more details or a reference to the agrobacterium growth and infiltration; for instance, what was the ‘appropriate’ OD600? Where the Agrobacterium suspensions stored for few hours before infiltration? Etc.

Reply: Done. We added more details and supplemented it at lines 200-202 in the revision. For agrobacterium concentration, exact OD600 value differed in different experiments and was shown in Figure legend.

Q6:  L 194 what is the 35S-Hc-Prom1-m12?

Reply: Those are twelve Hc-Pro mutants described in “Materials and Methods 2.4” and were described clearly in lines 190-193 and 205-206 in the revision.

Q7:  How was the success of the plasmid construction and protein expression (if they were full length for instance) of the BiFC assay verified?

Reply: We drew a flow chart to show BiFC principle (Supp. Fig. 4A), and we verified BiFC recombinant vector by PCR amplification and restriction endonuclease digestion (Supp. Fig. 4B, C). BiFC protein was visualized by confocal detection while Western blot is not applicable here as the expression should be too low to be detectable.

Q8:  Figures showing the BiFC assay in 5B are of poor quality. Only two cells show nuclei by DAPI, and the size of the nuclei seem overenlarged, suggesting that either the DAPI staining formed blobs or that the nuclei have been altered by the experimental conditions. Can the author provide better and more clear images, and can they address these concerns? Are there positive and negative controls included and indicated?

Reply: we provide better image here, please refer to new Fig.6B.

Q9:  Figure 1c and 1, and 3d: pictures are dark and it is hard to see the symptoms on the infected leaves, provide better images and a close up of parts of the laminar blade showing symptoms.

Reply: we improved the images here, please refer to new Fig.1C, D and Fig.4D.

Q10:  Figure 2b: is the healthy leaf on the bottom middle switched with the bottom right?

Reply:  Yes, it was switched. Thank the reviewer for careful reviewing. We corrected it now, please refer to new Fig.2B.

Q11:  Figure 6: why is the wt HC-Pro control not showing any silencing suppressor activity but some of its mutants are?

Reply: Here we want to distinguish the degree of silencing suppressor activity of Hc-Pro and its mutants, therefore we used low concentration of inoculum (OD600=0.1). At higher concentration, Hc-Pro would have some suppressor activity.

Reviewer 2 Report

Article: Synthesis of full-length cDNA infectious clones of Soybean mosaic virus and functional identification of a key amino acid in silencing suppressor Hc-Pro

Authors: Wenhua Bao Ting Yan, Xiaoyi Deng, and Hada Wuriyanghan

It was a pleasure reviewing the research article submitted to Viruses (viruses-867567). The manuscript describes the synthesis of full-length cDNA infectious clones of Soybean mosaic virus (SMV) and the identification of amino acids involved in the suppression of the host’s RNAi. The manuscript is informative and well organized. However, it lacks novelty. For instance, potyvirus infectious clones were constructed in the early 2000s. Similarly, it was also known about seed transmissibility of both SMW and alteration in CHSexpression.

Major comments:

The authors performed separate infection assays with SMW and GFP-SMW (Fig 1 and 2) in N. benthamiana and Soybean. These results show that both the constructs are infection. However, from Fig. 4A, it is clear that SMW induces stronger symptoms compared to GFP-SMW. Hence, a comparative molecular biology assay (Northern gel blot and RT-qPCR) is required to evaluate the efficiency of virus accumulation in SMW and GFP-SMW infected N. benthamiana and Soybean as well as all the 5 positive plants.

What is the reason for using different inoculation methods of N. benthamiana and Soybean? The soybean plants showed milder symptoms compared to N. benthamiana. Is that because agro-inoculation delivers more SMW than rub-inoculation? How SMW and GFP-SMW accumulated in N. benthamiana and Soybean when both are inoculated by the same method?

Fig 3. RT-PCR is not a quantitative assay whereas RT-qPCR (please changed qRT-PCR to RT-qPCR in the manuscript) is highly error-prone. Hence, performing a Northern gel blot assay for SMW and a housekeeping is widely accepted technique to support RT-qPCR data. Therefore, it is recommended to perform northern blot assay for all results where authors want to show the change in RNA level. Further, L429-430 states that “SMV-induced seed pigmentation was associated with the suppression of tissue-specific silencing of CHS gene expression”. However, Fig 3C demonstrates the over-expression of CHS mRNA in SMW inoculated plants compared to control.

Fig 5. Please explain details of using SD/-Try/-Leu medium, SD/-Try/-Leu/-His/X-α-Gal media, and selection of blue-colored colonies. It is advisable to present a flow chart of the experiment with positive and negative results. Although figure legend mentions positive and negative controls, the figure lacks the same (both 5A and 5B).

Fig 6. Provide a flow chart on Fig 6A and 6B with the positive and negative result so that readers could understand the experiment clearly. Although figure legend mentions positive and negative controls, the figure lacks the same (both 6B and 6D). For 6C, I would recommend performing Rt-qPCR and Northern gel blot for quantification.

L450: “Generally speaking, the host range of mild strains is narrower, while…. SMV infectious clones.” Although authors use mild strain in the present study, they found mild can infect a couple of plant species indicating it has a broad host range similar to severe strains. As a scientific research article, it is best to give reference to the statements such as “Generally speaking, the host range of mild strains is narrower, while that of severe strains is relatively wide”. Further, is this statement is based on disease symptom observation or by molecular diagnostics? This is very important because often mild strain infection not necessarily exhibit disease symptoms.

Minor comments:

  1. L18: recombination system respectively >> recombination system, respectively.
  2. Introduce the term before abbreviating. For instance, L23: Hc-Pro; L24: FRNK.
  3. L145-147: “The correct assembled…. Agroinfiltration”. Explain in detail how the correct assembled recombinant virus was examined.
  4. L164-175: provide details on agro-infiltration, names of legumes used, time points samples collected.
  5. L178: “…Full length…. cloned into...” Is it PCR amplified, or RT-PCR amplified?
  6. L181-183: “The twelve …., respectively”. Please verify with L362-364.
  7. L188-195: Provide details on where and how agrobacterium was grown, to what concentration agrobacterium was diluted, and what was the final concentration used for agro-infiltration.
  8. L216: … Briefly, Total RNA was…>> Briefly, total RNA was
  9. L219: … HiScriptIII 1st Strand cDNA..>> HiScriptIII 1st Strand cDNA
  10. L221: (Analytik jena)>> (Analytik Jena)
  11. L222: qRT-PCR>> RT-qPCR. Make similar changes in other places too.
  12. L215-225: Please mention how RNA quality and quantity were analyzed and the quantity of RNA was used for each quantification experiment (Rt-qPCR’ Northern blot).
  13. L313-335: Explain how plants were inoculated.
  14. L402: Host-virus interactions are the main topics in the field of plant virology. >> Host-virus interaction is one of the important topics in the field of plant virology.
  15. L334: Ref [54] is outdated at that place!
  16. L426: “SMV can be transmitted through seeds and causes seed mottle on the epidermis”. Provide reference.
  17. Please provide a conclusion of the manuscript.

Author Response

Editor’s comments:

Manuscript ID: viruses-867567

Title: Synthesis of full-length cDNA infectious clones of Soybean mosaic virus and functional identification of a key amino acid in silencing suppressor Hc-Pro

Please revise the manuscript according to the reviewers' comments and upload the revised file within 10 days. Use the version of your manuscript found at the above link for your revisions, as the editorial office may have made formatting changes to your original submission. Any revisions should be clearly highlighted, for example using the "Track Changes" function in Microsoft Word, so that changes are easily visible to the editors and reviewers. Please provide a cover letter to explain point-by-point the details of the revisions in the manuscript and your responses to the reviewers' comments. Please include in your rebuttal if you found it impossible to address certain comments. The revised version will be inspected by the editors and reviewers. Please detail the revisions that have been made, citing the line number and exact change, so that the editor can check the changes expeditiously. Simple statements like ‘done’ or ‘revised as requested’ will not be accepted unless the change is simply a typographical error.

Responses to editor: Overall, the editor’s and reviewer’s comments we received in last submission are very insightful and helpful. We really appreciate the comments and thoroughly revised the manuscript based on their comments.

We made substantial improvements on the manuscript (Manuscript ID: viruses-867567) by the following. 1)We detailed the methods and data. 2) We added some new data. 3) We revised the Figures. 4) We revised the writings. Followings are the reply to per comment.

Please note that we showed the line numbers to clearly tell where some revisions were made, while we realized that sometimes this line number changed a little (by one or two numbers) when creating the merged file.

Reviewer #2 (Comments for the Author):

General comments

Major comments:

Q1:  The authors performed separate infection assays with SMW and GFP-SMW (Fig 1 and 2) in N. benthamiana and Soybean. These results show that both the constructs are infection. However, from Fig. 4A, it is clear that SMW induces stronger symptoms compared to GFP-SMW. Hence, a comparative molecular biology assay (Northern gel blot and RT-qPCR) is required to evaluate the efficiency of virus accumulation in SMW and GFP-SMW infected N. benthamiana and Soybean as well as all the 5 positive plants.

Reply: We carried out RT-qPCR to evaluate virus accumulation (virus titer) in seven host plants upon infection of two different infectious clones (SMV, SMV-GFP) and the results were shown as new Fig. 5C.

Q2:  What is the reason for using different inoculation methods of N. benthamiana and Soybean? The soybean plants showed milder symptoms compared to N. benthamiana. Is that because agro-inoculation delivers more SMW than rub-inoculation? How SMW and GFP-SMW accumulated in N. benthamiana and Soybean when both are inoculated by the same method?

Reply: N. benthamiana is suitable for both agro-inoculation and rub-inoculation, while soybean is prone to rub-inoculation and agro-inculation is not easy to perform. We did not see any differences for N. benthamiana from agro- or rub- inoculations.

Q3:  Fig 3. RT-PCR is not a quantitative assay whereas RT-qPCR (please changed qRT-PCR to RT-qPCR in the manuscript) is highly error-prone. Hence, performing a Northern gel blot assay for SMW and a housekeeping is widely accepted technique to support RT-qPCR data. Therefore, it is recommended to perform northern blot assay for all results where authors want to show the change in RNA level. Further, L429-430 states that “SMV-induced seed pigmentation was associated with the suppression of tissue-specific silencing of CHS gene expression”. However, Fig 3C demonstrates the over-expression of CHS mRNA in SMW inoculated plants compared to control.

Reply: Thanks for the comments. 1) We changed qRT-PCR to RT-qPCR in this revision. 2) Fig. 3C is a quantitative assay via RT-qPCR, and Fig.3F is a qualitative assay by regular RT-PCR. We tried to detect CHS gene expression by Northern blot using Digoxin label while it was not successful as RNA isolated from seed sample was in low quantity. Also, isotope (32p) label is not available now. 3) SMV induces CHS expression as it suppresses silencing of CHS expression by endogenous siRNA, so our results are consistent with the statement in L429-430 (now L455-456 in this revision).

Q4:  Fig 5. Please explain details of using SD/-Try/-Leu medium, SD/-Try/-Leu/-His/X-α-Gal media, and selection of blue-colored colonies. It is advisable to present a flow chart of the experiment with positive and negative results. Although figure legend mentions positive and negative controls, the figure lacks the same (both 5A and 5B).

Reply: We drew a flow chart to explain the experimental details and were provided as Supp. Fig. 3.

Q5:  Fig 6. Provide a flow chart on Fig 6A and 6B with the positive and negative result so that readers could understand the experiment clearly. Although figure legend mentions positive and negative controls, the figure lacks the same (both 6B and 6D). For 6C, I would recommend performing Rt-qPCR and Northern gel blot for quantification.

Reply: We drew a flow chart to explain the experimental details (mutation sites) and were provided as Fig. 7A. For 6C, we did Rt-qPCR for quantification using GFP primers and the results were shown as Fig.7D here.

Q6:  L450: “Generally speaking, the host range of mild strains is narrower, while…. SMV infectious clones.” Although authors use mild strain in the present study, they found mild can infect a couple of plant species indicating it has a broad host range similar to severe strains. As a scientific research article, it is best to give reference to the statements such as “Generally speaking, the host range of mild strains is narrower, while that of severe strains is relatively wide”. Further, is this statement is based on disease symptom observation or by molecular diagnostics? This is very important because often mild strain infection not necessarily exhibit disease symptoms.

 Reply: we declared that this statement was inappropriate and therefore was deleted.

Minor comments:

Q7:  L18: recombination system respectively >> recombination system, respectively.

Reply: Done. Corrected as recommended on line 19.

 Q8:  Introduce the term before abbreviating. For instance, L23: Hc-Pro; L24: FRNK.

Reply: Done. The full name was shown on line 24 and line 26, respectively .

 Q9:  L145-147: “The correct assembled…. Agroinfiltration”. Explain in detail how the correct assembled recombinant virus was examined.

Reply: Done. The recombinant virus was verified by restriction endonuclease digestion, PCR amplification and Sanger sequencing. This was explained on lines 146-147.

 Q10:  L164-175: provide details on agro-infiltration, names of legumes used, time points samples collected.

Reply: Done. Explained in detail as recommended on lines 166-181.

 Q11:  L178: “…Full length…. cloned into...” Is it PCR amplified, or RT-PCR amplified?

Reply: Done. It is PCR amplified from SMV infectious clone DNA and was clarified on line 187.

 Q12:  L181-183: “The twelve …., respectively”. Please verify with L362-364.

Reply: Done. We checked them and verified them on lines 191-193 and 402-404.

 Q13:  L188-195: Provide details on where and how agrobacterium was grown, to what concentration agrobacterium was diluted, and what was the final concentration used for agro-infiltration.

Reply: Done. Explained in detail as recommended on lines 198-206.

 Q14:  L216: … Briefly, Total RNA was…>> Briefly, total RNA was

Reply: Done. Corrected as recommended on line 226.

 Q15:  L219: … HiScriptIII 1st Strand cDNA..>> HiScriptIII 1st Strand cDNA

Reply: Done. Corrected as recommended on line 231.

 Q16:  L221: (Analytik jena)>> (Analytik Jena)

Reply: Done. Corrected as recommended on line 233.

 Q17:  L222: qRT-PCR>> RT-qPCR. Make similar changes in other places too.

Reply: Done. Corrected as recommended on line 233 and in whole manuscript.

 Q18:  L215-225: Please mention how RNA quality and quantity were analyzed and the quantity of RNA was used for each quantification experiment (Rt-qPCR’ Northern blot).

Reply: The detail was provided on lines 228-230.

 Q19:  L313-335: Explain how plants were inoculated.

Reply: The detail was added on lines 335-363.

 Q20:  L402: Host-virus interactions are the main topics in the field of plant virology. >> Host-virus interaction is one of the important topics in the field of plant virology.

Reply: Done on line 428.

 Q21:  L334: Ref [54] is outdated at that place!

Reply: We cited new reference here as ref55 on line 430.

 Q22:  L426: “SMV can be transmitted through seeds and causes seed mottle on the epidermis”. Provide reference.

Reply: Done. Provide references 14 and 15 on line 452.

 Q23:  Please provide a conclusion of the manuscript.

Reply: Done. provided on lines 504-515.

Round 2

Reviewer 1 Report

Thank you for adding the extra experiments and data and for clarifying my doubts. I do not have any further comment.

Author Response

Thanks the reviewer for reviewing this work.

Reviewer 2 Report

Revision greatly improved the quality of the article.

Author Response

Thank the reviewer for reviewing the manuscript.